# Primary health care utilization in the first year after arrival by refugee sponsorship model in Ontario, Canada: A population-based cohort study

Susitha Wanigaratne [1,2,3]*, Jennifer Rayner[4,5], Richard H. Glazier[2,4,6,7,8], Therese A. Stukel[2,4,9], Hong Lu[2], Sima Gandhi[2], Natasha R. Saunders[1,2,3,9,10,11], Michaela Hynie [12], Anja Kilibarda[13], Astrid Guttmann[1,2,3,9,10,11,14]

1 Edwin S.H. Leong Centre for Healthy Children, University of Toronto, Toronto, Canada, 2 ICES, Toronto, Canada, 3 Child Health Evaluative Sciences, SickKids Research Institute, Toronto, Canada, 4 Institute of Health Policy, Management and Evaluation, The University of Toronto, Toronto, Canada, 5 Alliance for Healthier Communities, Toronto, Canada, 6 Department of Family and Community Medicine, University of Toronto, Toronto, Canada, 7 Department of Family and Community Medicine, St. Michael's Hospital, Toronto, Canada, 8 MAP Centre for Urban Health Solutions, St. Michael's Hospital, Toronto, Canada, 9 Temerty Faculty of Medicine, University of Toronto, Toronto, Canada, 10 Division of Paediatric Medicine, the Hospital for Sick Children, Toronto, Canada, 11 Department of Paediatrics, University of Toronto, Toronto, Canada, 12 Department of Psychology/Centre for Refugee Studies, York University, Toronto, Canada, 13 Columbia University Department of Political Science, New York City, New York, United States of America, 14 Dalla Lana School of Public Health, University of Toronto, Toronto, Canada

* susitha.wanigaratne@sickkids.ca

## Abstract

### Background

Canada's approach to refugee resettlement includes government sponsorship, a pioneering private sponsorship model and a third blended approach. Refugees are selected and supported differently in each approach including healthcare navigation. Little is known about how well private sponsors facilitate primary care navigation and whether this changed during the large-scale 2015 Syrian resettlement initiative characterized by civic and healthcare systems engagement.

### Methods and findings

Population-based cohort study of resettled refugees arriving in Ontario between April 1, 2008 and March 31, 2017, with one-year follow-up, using linked health and demographic administrative databases. We evaluated associations of resettlement model (GARs, Privately Sponsored Refugees [PSRs], and Blended-Visa Office Referred [BVORs]) by era of arrival (pre-Syrian and Syrian era) and by country cohort, on measures of primary care (PC) navigation using adjusted Cox proportional hazards and logistic regression. There were 34,591 (pre-Syrian) and 24,757 (Syrian era) resettled refugees, approximately half of whom were GARs. Compared with the reference group pre-Syrian era PSRs, Syrian PSRs had slightly earlier PC visits (mean = 116 days [SD = 90]) (adjusted hazard ratios [aHR] = 1.19, 95% CI 1.14–1.23). Syrian GARs (mean = 72 days [SD = 65]) and BVORs (mean = 73 days

**Data Availability Statement:** The data sets from this study are held securely in coded form at ICES. Data-sharing agreements prohibit ICES from making the data sets publicly available, but access may be granted to those who meet pre-specified criteria for confidential access, available at https://www.ices.on.ca/das. The data set creation plan is included in Appendix Table 2 and the underlying analytic code is available from the authors upon request or by emailing das@ices.on.ca, understanding that the programs may rely upon coding templates or macros that are unique to ICES.

**Funding:** Principal Investigator AG was funded by the Canadian Institutes of Health Research (https://cihr-irsc.gc.ca/e/193.html) - grant PJT 155917. The funders had no role in study design, data collection and analysis, decision to publish, or preparation of the manuscript.

**Competing interests:** The authors have declared that no competing interests exist.

[SD = 76]) had their first PC visit sooner than pre-Syrian era PSRs (mean = 149 days [SD = 86]), with respective aHRs 2.27, 95% CI 2.19–2.35 and 1.89, 95% CI 1.79–1.99. Compared to pre-Syrian PSRs, Syrian GARs and BVORs had much greater odds of a CHC visit (adjusted odds ratios 14.69, 95% CI 12.98–16.63 and 14.08, 95% 12.05–16.44 respectively) and Syrian PSRs had twice the odds of a CHC visit.

## Conclusions

Less timely primary care and lower odds of a CHC visit among PSRs in the first year may be attributed to selection factors and gaps in sponsors' knowledge of healthcare navigation. Improved primary care navigation outcomes in the Syrian era suggests successful health systems engagement.

## Background

Globally, the population of forcibly displaced people has increased by 50% in the past decade resulting in over 26 million refugees in 2019 [1]. While the need for refugee protection has increased, resettlement funding and opportunities have not kept apace [2]. Since 1959 Canada has resettled nearly 700,000 refugees and in 2018 accepted the highest number of refugees per capita and overall amongst 27 high-income countries committed to the United Nations High Commissioner for Refugees (UNHCR) resettlement program. In response to the Syrian migrant crisis, starting in late 2015 Canada accepted over 40,000 refugees within an 18-month period. The Syrian response was characterized by unprecedented mobilization and engagement of provincial governments [3] (including their health systems [4, 5]), the settlement sector [6] and civil society [7, 8]. Canada's refugee resettlement program is characterized by government sponsorship, a pioneering private sponsorship program [9] in which private citizens are responsible for settlement support for the first year, and a third model blending the two. Given the global demand for resettlement, the Global Refugee Sponsorship Initiative [9] has helped spread Canada's private sponsorship program, specifically the approach where refugees are referred by the UNHCR or migration agency and private sponsors are unknown to the refugees they sponsor. Such programs have been implemented in several other countries including the United Kingdom, New Zealand and most recently to the United States [10, 11].

Canada's three resettlement programs provide permanent residence upon arrival and resettlement supports (i.e. financial, instrumental, informational, emotional) [12] for at least the first year. Table 1 summarizes resettlement supports and health care coverage available to refugee resettlement groups. Refugees with acute medical needs or those who have experienced violence, trauma or other adversities are referred by the UNHCR for mostly government assistance [13, 14] or blended sponsorship [15]. Government-assisted refugees are convention refugees who receive financial assistance, formal immigration resettlement support and case management in the first year of arrival through the Canadian government's Resettlement Assistance Program (RAP). The blended visa office-referred program matches convention refugees with private sponsors who provide resettlement support while financial assistance is provided for 6 months each by the RAP and sponsors [4, 12]. Privately sponsored refugees must either be a convention refugee or a member of the country of asylum class and are often referred by friends or family in Canada [16, 17]. They are supported entirely by sponsors (i.e., religious, humanitarian or private citizen group) during their first year in Canada. Navigating

**Table 1. Resettled refugee streams–selection criteria and responsibility for provision of resettlement support and health care coverage in the first year after immigration.**

| Refugee resettlement group | Government Assisted Refugees | Blended Visa Office Referred Refugees | Privately Sponsored Refugees (Sponsorship Agreement Holder[2], Groups of Five, Community Sponsored) |
|---|---|---|---|
| **Referred by** | UNHCR[1] or migration agency | | Private sponsors, UNHCR[1] or migration agency, matched to sponsors[3] |
| **Income support** (first yr) | Federal govt[4] | 50/50 Federal govt[4] & Private sponsors | Private sponsors |
| **Resettlement support/navigation** (first yr; may continue > first yr w/ private sponsors) | Federal govt[4] | Private sponsors | Private sponsors |
| **Healthcare coverage** (first yr and after) | Provincial govt | Provincial govt | Provincial govt |
| **Supplemental health benefits[5]** (first yr) | Federal govt | Federal govt | Federal govt |

[1] Selected based on one or more resettlement criteria [20] including medical need, survivors of violence and trauma, women at risk and heightened vulnerability and urgency of resettlement need.

[2] incorporated groups including religious, ethnic, community-based and settlement service organizations with ongoing sponsorship agreements with the federal government. These groups are involved in "named" or "family-linked" sponsorships–having a personal connection to the group or are the family of refugees already living in Canada.

[3] After 2012, Group of 5 and Community Sponsored privately sponsored refugees shifted to more UNHCR referrals but selection by private sponsors was still possible.

[4] Through the Resettlement Assistance Program (RAP).

[5] Provided through the Interim Federal Health Program administered by Blue Cross. Covers some dental and vision care as well as prescription drug coverage.

the healthcare system is one important early resettlement process with consequences for many long-term health and social outcomes, yet little is known about how well primary healthcare needs are met across refugee resettlement models. A criticism of private sponsorship models (for privately sponsored and blended-visa office referred refugees), is the lack of mandatory resettlement training for sponsors [18], which may be particularly important for health system navigation. Private sponsors have suggested that next to housing, accessing healthcare services is the biggest early resettlement challenge[19].

In Ontario, all resettled refugees are eligible for publicly funded provincial healthcare insurance on arrival. For recent refugees who often have unmet healthcare needs, primary care is particularly important. Primary care is central to screening, prevention and treatment and is the gateway to specialized medical care. In Ontario, there are distinct models of primary care delivery with about 75% of Ontario's population belonging to a Patient Enrollment Model (PEM), often associated with comprehensive primary care [21, 22]. Community Health Centres (CHCs) include a specialized model of primary care integrated with social services and community programming [23] serving Ontarians facing barriers. Several CHCs are specifically oriented to immigrant health, including formal settlement services in-house and access to interpretation services making it easier for immigrants to connect to and navigate primary health care services. During the mass resettlement of Syrian refugees in 2015, CHCs in Ontario mobilized as a sector to meet the increased demand for services [6].

This study leverages a population-based provincial data repository with linked immigration and health administrative data for Ontario, Canada. Focusing on resettled refugees in the first year after arrival, the objectives of our study were to: 1) describe time to first primary care visit and use of CHCs by sponsorship model within the pre-Syrian (April 1, 2008 to October 31, 2015) and Syrian (November 1, 2015 to March 31, 2017) settlement eras, 2) test for differences in these measures of healthcare navigation across a combination of sponsorship model, settlement era, and across distinct country cohorts of refugees. Our first hypothesis is that government-assisted refugees will attend their first primary care visit earlier and be more likely to have a CHC visit than blended-visa office referred and privately sponsored refugees. This is

related to: i) differential selection processes (i.e., government-assisted and blended-visa office referred refugees are selected by the UNHCR based on urgency and vulnerability and thus may require more healthcare), and ii) private sponsor's differing knowledge of primary health care navigation. Our second hypothesis is that refugees resettled during the Syrian era, particularly those from Syria, compared to those resettled in the pre-Syrian era, will have more timely primary care access related to the increased civic and government engagement in response to the Syrian migrant crisis [3, 5–8].

## Materials and methods

### Study design and cohort

We conducted a population-based cohort study of all resettled refugees who landed in Ontario between April 1, 2008 and March 31, 2017 (N = 59, 701). We followed the cohort for one year after arrival or until death. We excluded those without a valid Ontario health card number (N = 329, 0.55%) and a small number of records for individuals who had more than one arrival record before April 1, 2008 (N = 24, 0.04%) to limit our sample to those new to the Ontario healthcare system. See S1 Fig for database linkage and cohort selection flowchart. "Protected Persons", also known as successful "refugee claimants" or "asylum seekers", were not included since they are not sponsored by the government or a private group and have a different migration pathway to Canada [24]. We followed relevant reporting guidelines [25, 26].

### Data sources

We used linked demographic and healthcare administrative databases available at ICES (see S1 Table for summary of all data sources and S2 Table for study variable details). ICES is an independent, non-profit research institute whose legal status under Ontario's health information privacy law allows it to collect and analyze health care and demographic data, without consent, for health system evaluation. Resettled refugees were identified from the ICES data repository which links the Ontario portion of the Immigration, Refugees and Citizenship Canada (IRCC) Permanent Resident Database to the healthcare registry. About 92% of refugees (resettled refugees and protected persons) are linked and included in the ICES data repository (see S1 Fig for linkage information). The healthcare registry includes demographic information and vital statistics on all Ontario residents who have been eligible for publicly funded healthcare and have received a health card number since 1990. This single-payer system funds access to most medically necessary health-care services. There are few differences in characteristics between refugees linked to the healthcare registry compared to those unlinked [27], suggesting minimal linkage bias. The study cohort was then deterministically linked to healthcare databases using unique encoded identifiers.

### Primary exposures

The two primary exposures of interest were refugee resettlement model and era of landing. We grouped blended-visa office referred refugees with privately sponsored refugees during the pre-Syrian era due to small numbers of the former. We categorized era of landing into a binary variable: pre-Syrian (April 1, 2008 to October 31, 2015) and Syrian (November 1, 2015 to March 31, 2017) eras. The pre-Syrian era's large resettled refugee population served as a comparison for those landing in the Syrian-era which marked the start of the Government of Canada's commitment to resettle Syrian refugees who had fled to neighbouring countries [28]. Syrian and non-Syrian refugees landing in the Syrian era were examined separately. Pre-Syrian era privately sponsored refugees were chosen as the reference group.

### Secondary exposures

For secondary analyses, we examined refugee country cohorts for which there were explicit Canadian resettlement commitments (Syria [29], Bhutan [30], Myanmar [31], Iraq [32]) or had large population sizes (Afghanistan, Iran, Somalia, Eritrea, Democratic Republic of Congo, Ethiopia). We grouped refugees from "all other African countries" and from "all other countries" (reference).

### Primary outcomes

Primary outcomes were 1) time in days from arrival in Canada to first primary care visit and 2) any CHC visit in the first year after landing. We identified primary care visits to a family physician/general practitioner (GP), pediatrician, or nurse practitioner using previously validated physician billing fee and diagnosis codes and CHC electronic medical records [33, 34].

### Secondary outcomes

Secondary outcomes included: i) primary care visit in the first two months and at the end of the first year, ii) primary care enrollment model or provider at the end of the first year using previously validated methods [33] and, iii) "any major morbidity" measured using the Johns Hopkins ACG® System Aggregated Diagnosis Groups (ADGs) case-mix adjustment system (version 10).

### Other baseline characteristics and covariates

These included age at the time of arrival, sex, rural/urban residence and census area-level material deprivation [35]. Immigration characteristics included world region of origin, secondary migration, family status and Canadian language ability. For adults ($\geq$18 years) we included marital status and the highest education level (in those $\geq$25 years old). We also estimated drive time to the nearest CHC in minutes as a measure of accessibility.

### Statistical analysis

The dataset creation plan outlining the methodology and analyses can be found in S3–S5 Tables. Analyses were all pre-specified. In the pre-Syrian era, blended-visa office referred refugees were aggregated with government-sponsored refugees due to small numbers. In each era, we estimated standardized differences comparing government-assisted refugees to privately sponsored refugees for all baseline characteristics. Standardized differences >0.1 were deemed important [36]. We tested differences between sponsorship models across healthcare use outcomes for the pre-Syrian and Syrian era using $\chi^2$ tests for categorical variables and $t$-tests or ANOVA tests for continuous variables ($p<0.05$). Given the large population sizes we acknowledge that small differences in outcomes will be significantly different therefore our interpretation is informed by clinical rather than statistical significance.

In all regression analyses, a small number of refugees living in rural areas or with missing rurality were excluded due to differences in access to health services in rural areas. For primary outcomes we examined the *a priori* hypothesis of an interaction between settlement model and era. Since the interaction term for both primary outcomes was statistically significant, we re-parameterized the exposure to account jointly for both era and sponsorship model effects. We used Kaplan-Meier survival curves to plot the proportion with a primary care visit over time by settlement model and era. We used multi-variable Cox proportional hazards models to test the association between time to first primary care visit and sponsorship model and era of landing, with Syrian and non-Syrian refugees examined separately in the Syria era. We used multi-

variable logistic regression to model the odds of a CHC visit in the first year after landing. Secondary analyses by country of origin followed the same analytic plan but also adjusted for year of landing and used a referent group of privately sponsored refugees from all other countries not separately specified. We recognized that if CHC visits are prevalent, logistic regression models will overestimate relative risks therefore we report odds rather than risks or probability.

**Subgroup analyses.** Since marital status and education are measured at arrival and relevant only for adult health outcomes, we adjusted for these variables in age restricted ($\geq$25) models.

### Ethics approval

The use of data in this project was authorized under section 45 of Ontario's Personal Health Information Protection Act, which does not require review by a Research Ethics Board. The data were fully anonymized and used an encrypted unique identifier to link across databases prior to analysis.

## Results

### Baseline characteristics

In both eras, privately sponsored refugees were generally older, single, had higher education, were more likely to speak English, and less likely to arrive with a child or other dependent compared to government-assisted and blended-visa office referred refugees (Table 2). In the Syrian era, government-assisted refugees were more likely to live in the most materially deprived neighborhoods (61.9%) compared to privately sponsored refugees (55.8%) while blended-visa office referred refugees were less likely (49.6%) with similar proportions residing in less deprived neighborhoods as privately sponsored refugees. Government-assisted and blended-visa office referred refugees in the Syrian era had lower levels of education than in the pre-Syrian era which was the opposite for privately sponsored refugees. Across all sponsorship models, most refugees were citizens of Africa and the Middle East with larger proportions of government-assisted and blended-visa office referred refugees from Syria than privately sponsored refugees in the Syrian era. In both eras government-assisted refugees were much more likely to live within 3 km drive time to a CHC than privately sponsored refugees.

### Primary care and CHC healthcare utilization

Between 77% and 95% of all resettled refugees in both the pre-Syrian and Syrian eras had at least one primary care visit in the first year of resettlement; however across sponsorship groups in both eras there was variation in both primary and most secondary outcomes (Table 3). Privately sponsored refugees were less likely to have a major morbidity compared to government-assisted and blended-visa office referred refugees, particularly in the pre-Syrian era, and took longer to have a primary care visit compared to government-assisted and blended-visa office referred refugees (mean days–pre-Syrian era: privately sponsored refugees = 149.4, government-assisted refugees = 78.0; Syrian era: privately sponsored refugees = 115.8, government-assisted refugees = 72.3, blended-visa office referred refugees = 73.0). Privately sponsored refugees were also less likely to have a primary care visit in the first year in both eras and had a lower average number of primary care visits per person compared to government-assisted (in both eras) and blended-visa office referred refugees (Syrian era only). The proportion of all resettled refugees with a CHC visit in the first year doubled in the Syrian (17.8%) versus pre-Syrian (8.2%) era. The majority of refugees, regardless of sponsorship model, were either

**Table 2. Sociodemographic characteristics of resettled refugees who landed in Ontario between April 1, 2008 and March 31, 2017, by era of landing and sponsorship model, N (% of column unless otherwise indicated).**

| Sponsorship Model | Pre-Syrian era–April 1, 2008 to October 31, 2015 | | | Syrian era–November 1, 2015 to March 31, 2017 | | | | |
|---|---|---|---|---|---|---|---|---|
| | Government-assisted refugees (GARs) | Privately sponsored refugees (PSRs)[1] | Standardized Difference | Government-assisted refugees (GARs) | Blended Visa Office-referred refugees (BVORs) | Privately sponsored refugees (PSRs) | Standardized Difference GARs vs. PSRs | Standardized Difference BVORs vs. PSRs |
| **Cohort size, N** | 17,623 | 16,968 | | 12,051 | 2,695 | 10,011 | | |
| **Age at landing date** | | | | | | | | |
| Mean ± SD | 25.9 ± 17.9 | 28.5 ± 17.8 | 0.14 | 20.02 ± 16.3 | 19.49 ± 15.7 | 27.86 ± 18.5 | 0.45 | 0.49 |
| Median (IQR) | 23 (12–38) | 26 (15–40) | 0.16 | 15 (6–32) | 14 (6–32) | 27 (12–40) | 0.44 | 0.47 |
| **Age group** | | | | | | | | |
| 0–5 | 1,964 (11.1%) | 1,532 (9.0%) | 0.07 | 2,580 (21.4%) | 574 (21.3%) | 1,197 (12.0%) | 0.26 | 0.25 |
| 6–11 | 2,435 (13.8%) | 1,725 (10.2%) | 0.11 | 2,551 (21.2%) | 600 (22.3%) | 1,256 (12.5%) | 0.23 | 0.26 |
| 12–17 | 2,432 (13.8%) | 1,889 (11.1%) | 0.08 | 1,433 (11.9%) | 320 (11.9%) | 962 (9.6%) | 0.07 | 0.07 |
| 18–30 | 4,366 (24.8%) | 4,839 (28.5%) | 0.08 | 2,093 (17.4%) | 445 (16.5%) | 2,297 (22.9%) | 0.14 | 0.16 |
| 31–45 | 3,788 (21.5%) | 3,954 (23.3%) | 0.04 | 2,501 (20.8%) | 596 (22.1%) | 2,558 (25.6%) | 0.11 | 0.08 |
| 46–65 | 2,097 (11.9%) | 2,486 (14.7%) | 0.08 | 759 (6.3%) | 144 (5.3%) | 1,401 (14.0%) | 0.26 | 0.3 |
| 65+ | 541 (3.1%) | 543 (3.2%) | 0.01 | 134 (1.1%) | 16 (0.6%) | 340 (3.4%) | 0.15 | 0.2 |
| **Sex** | | | | | | | | |
| Female | 8,957 (50.8%) | 8,157 (48.1%) | 0.06 | 5,880 (48.8%) | 1,329 (49.3%) | 4,773 (47.7%) | 0.02 | 0.03 |
| Male | 8,666 (49.2%) | 8,811 (51.9%) | 0.06 | 6,171 (51.2%) | 1,366 (50.7%) | 5,238 (52.3%) | 0.02 | 0.03 |
| **Neighborhood material deprivation quintile** | | | | | | | | |
| 1 (Least deprived) | 162 (0.9%) | 507 (3.0%) | 0.15 | 84 (0.7%) | 160 (5.9%) | 462 (4.6%) | 0.25 | 0.06 |
| 2 | 275 (1.6%) | 1,008 (5.9%) | 0.23 | 216 (1.8%) | 273 (10.1%) | 861 (8.6%) | 0.31 | 0.05 |
| 3 | 956 (5.4%) | 1,542 (9.1%) | 0.14 | 531 (4.4%) | 357 (13.2%) | 1,146 (11.4%) | 0.26 | 0.05 |
| 4 | 5,174 (29.4%) | 2,918 (17.2%) | 0.29 | 3,764 (31.2%) | 567 (21.0%) | 1,959 (19.6%) | 0.27 | 0.04 |
| 5 (Most deprived)[2] | 11,056 (62.7%) | 10,993 (64.8%) | 0.04 | 7,456 (61.9%) | 1,338 (49.6%) | 5,583 (55.8%) | 0.12 | 0.12 |
| **Rurality** | | | | | | | | |
| Rural | 7 (0.0%) | 47 (0.3%) | 0.06 | 6 (0.0%) | * | 75 (0.7%) | 0.11 | 0.39 |
| Urban | 17,552 (99.6%) | 16,790 (99.0%) | 0.08 | 12,035 (99.9%) | 2,452 (91.0%) | 9,832 (98.2%) | 0.17 | 0.32 |
| Missing | 64 (0.4%) | 131 (0.8%) | 0.05 | 10 (0.1%) | * | 104 (1.0%) | 0.13 | 0.12 |
| **Canadian language ability at arrival** | | | | | | | | |
| Bilingual | 250 (1.4%) | 225 (1.3%) | 0.01 | 26 (0.2%) | 7 (0.3%) | 31 (0.3%) | 0.02 | 0.01 |
| English | 3,770 (21.4%) | 5,822 (34.3%) | 0.29 | 1,587 (13.2%) | 766 (28.4%) | 5,316 (53.1%) | 0.94 | 0.52 |
| French | 400 (2.3%) | 169 (1.0%) | 0.1 | 100 (0.8%) | 12 (0.4%) | 101 (1.0%) | 0.02 | 0.07 |
| None[3] | 13,203 (74.9%) | 10,752 (63.4%) | 0.25 | 10,338 (85.8%) | 1,910 (70.9%) | 4,563 (45.6%) | 0.93 | 0.53 |
| **World region[4]** | | | | | | | | |
| Africa & Middle East | 13,352 (75.8%) | 12,923 (76.2%) | 0.01 | 11,563 (96.0%) | 2,585 (95.9%) | 8,787 (87.8%) | 0.3 | 0.3 |
| Americas | 354 (2.0%) | 124 (0.7%) | 0.11 | 30 (0.2%) | 18 (0.7%) | 6 (0.1%) | 0.05 | 0.1 |
| Asia & Pacific | 3,634 (20.6%) | 3,554 (20.9%) | 0.01 | 389 (3.2%) | 83 (3.1%) | 955 (9.5%) | 0.26 | 0.27 |
| Europe | 183 (1.0%) | 31 (0.2%) | 0.11 | 8 (0.1%) | * | 12 (0.1%) | 0.02 | |
| Stateless | 95 (0.5%) | 333 (2.0%) | 0.13 | 48 (0.4%) | * | 234 (2.3%) | 0.00 | |
| USA | * | * | | 0 (0.0%) | 0 (0.0%) | 7 (0.1%) | 0.17 | 0.04 |
| Not stated | * | * | | 13 (0.1%) | * | 10 (0.1%) | 0.04 | |
| **Secondary Migration** | | | | | | | | |
| Yes | 7,570 (43.0%) | 5,823 (34.3%) | 0.18 | 11,788 (97.8%) | 2,682 (99.5%) | 9,611 (96.0%) | 0.10 | 0.24 |

*(Continued)*

**Table 2.** (Continued)

| Sponsorship Model | Pre-Syrian era–April 1, 2008 to October 31, 2015 | | | Syrian era–November 1, 2015 to March 31, 2017 | | | | |
|---|---|---|---|---|---|---|---|---|
| | Government-assisted refugees (GARs) | Privately sponsored refugees (PSRs)[1] | Standardized Difference | Government-assisted refugees (GARs) | Blended Visa Office-referred refugees (BVORs) | Privately sponsored refugees (PSRs) | Standardized Difference GARs vs. PSRs | Standardized Difference BVORs vs. PSRs |
| **Marital Status at arrival (≥18 years)** | | | | | | | | |
| Single | 3,942 (36.5%) | 5,162 (43.7%) | 0.15 | 1,032 (18.8%) | 218 (18.2%) | 2,185 (33.1%) | 0.33 | 0.35 |
| Married | 5,686 (52.7%) | 5,912 (50.0%) | 0.05 | 4,185 (76.3%) | 915 (76.2%) | 4,012 (60.8%) | 0.34 | 0.34 |
| Separated, divorced, or widowed[3] | 1,164 (10.8%) | 748 (6.3%) | 0.16 | 270 (4.9%) | 68 (5.7%) | 399 (6.0%) | 0.05 | 0.02 |
| **Highest education level at arrival (≥25 years)** | | | | | | | | |
| ≤ secondary[3] | 6,485 (78.3%) | 6,382 (69.8%) | 0.19 | 3,916 (87.3%) | 894 (90.2%) | 3,178 (57.9%) | 0.7 | 0.79 |
| Trade, Diploma, some Uni. | 734 (8.9%) | 1,443 (15.8%) | 0.21 | 317 (7.1%) | 51 (5.1%) | 994 (18.1%) | 0.34 | 0.41 |
| ≥ Bachelor's | 1,065 (12.9%) | 1,321 (14.4%) | 0.05 | 251 (5.6%) | 46 (4.6%) | 1,315 (24.0%) | 0.54 | 0.57 |
| **Country cohort[5]** | | | | | | | | |
| Afghanistan | 1,135 (6.4%) | 2,405 (14.2%) | 0.26 | 70 (0.6%) | * | 736 (7.4%) | 0.35 | 0.4 |
| Bhutan | 1,098 (6.2%) | 0 (0.0%) | 0.36 | 117 (1.0%) | * | 0 (0.0%) | 0.14 | 0.04 |
| Congo | 925 (5.2%) | 222 (1.3%) | 0.22 | 331 (2.7%) | 47 (1.7%) | 48 (0.5%) | 0.18 | 0.12 |
| Eritrea | 283 (1.6%) | 1,223 (7.2%) | 0.28 | 100 (0.8%) | 89 (3.3%) | 995 (9.9%) | 0.41 | 0.27 |
| Ethiopia | 390 (2.2%) | 690 (4.1%) | 0.11 | 60 (0.5%) | 7 (0.3%) | 93 (0.9%) | 0.05 | 0.09 |
| Iran | 1,692 (9.6%) | 184 (1.1%) | 0.39 | 143 (1.2%) | 39 (1.4%) | 20 (0.2%) | 0.12 | 0.14 |
| Iraq | 7,186 (40.8%) | 8,463 (49.9%) | 0.18 | 753 (6.2%) | 118 (4.4%) | 874 (8.7%) | 0.09 | 0.18 |
| Myanmar | 1,016 (5.8%) | 217 (1.3%) | 0.25 | 121 (1.0%) | 77 (2.9%) | 15 (0.1%) | 0.11 | 0.22 |
| Somalia | 1,761 (10.0%) | 904 (5.3%) | 0.18 | 174 (1.4%) | * | 154 (1.5%) | 0.01 | 0.17 |
| Syria | N/A | N/A | | 9,747 (80.9%) | 2,201 (81.7%) | 6,542 (65.3%) | 0.36 | 0.38 |
| Other African | 725 (4.1%) | 435 (2.6%) | 0.09 | 231 (1.9%) | 83 (3.1%) | 150 (1.5%) | 0.03 | 0.11 |
| Other countries | 1,412 (8.0%) | 2,225 (13.1%) | 0.17 | 204 (1.7%) | 30 (1.1%) | 384 (3.8%) | 0.13 | 0.18 |
| **Family status** | | | | | | | | |
| Principal applicant | 7,212 (40.9%) | 7,540 (44.4%) | 0.07 | 3,222 (26.7%) | 694 (25.8%) | 4,351 (43.5%) | 0.36 | 0.38 |
| Spouse or common-law partner | 2,616 (14.8%) | 2,850 (16.8%) | 0.05 | 2,035 (16.9%) | 450 (16.7%) | 1,941 (19.4%) | 0.06 | 0.07 |
| Child or other dependent | 7,795 (44.2%) | 6,578 (38.8%) | 0.11 | 6,794 (56.4%) | 1,551 (57.6%) | 3,719 (37.1%) | 0.39 | 0.42 |
| **Drive Time to CHC (minutes)** | | | | | | | | |
| Missing | 72 (0.4%) | 131 (0.8%) | 0.05 | 9 (0.1%) | * | 105 (1.0%) | 0.13 | |
| ≤3 | 6,536 (37.1%) | 2,243 (13.2%) | 0.57 | 5,981 (49.6%) | * | 1,077 (10.8%) | 0.93 | |
| 3–10 | 8,543 (48.5%) | 10,231 (60.3%) | 0.24 | 4,614 (38.3%) | 1,124 (41.7%) | 5,436 (54.3%) | 0.33 | 0.25 |
| >10 | 2,472 (14.0%) | 4,363 (25.7%) | 0.3 | 1,447 (12.0%) | 1,060 (39.3%) | 3,393 (33.9%) | 0.54 | 0.11 |

[1] includes BVORs due to small counts.

[2] Includes those with suppressed deprivation data.

[3] Includes missing data due to small numbers (cell sizes <6).

[4] World regions were assigned based on source country (country of citizenship).

[5] Country cohorts represent groups of refugees fleeing recent conflicts. In some cases, a stated federal government commitment to resettle a target number of individuals was made. These groups were constructed based on CIC/IRCC Special Program indicators and/or source country.

* Data suppressed to reduce the risk of re-identification (for cell sizes <6).

**Table 3. Primary health care use and morbidity in the first year among resettled refugees who landed in Ontario between April 1, 2008 and March 31, 2017 by era of landing and sponsorship model.**

| | Pre-Syrian era–April 1, 2008 to October 31, 2015 | | | Syrian era–November 1, 2015 to March 31, 2017 | | | |
|---|---|---|---|---|---|---|---|
| Sponsorship Model | Government-assisted refugees (GARs) | Privately sponsored refugees (PSRs)[1] | | Government-assisted refugees (GARs) | Blended Visa Office-referred refugees (BVORs) | Privately sponsored refugees (PSRs) | |
| Cohort size, N | 17,623 | 16,968 | | 12,051 | 2,695 | 10,011 | |
| **Visits to Primary Care and Community Health Centres (CHCs)** | | | | | | | |
| **Time (days) from Landing date to first primary care healthcare contact, year 1[2]** | | | | | | | |
| Mean ± SD | 78.0 ± 76.7 | 149.4 ± 86.2 | [8] | 72.3 ± 64.8 | 73.0 ± 76.2 | 115.8 ± 90.4 | [8] |
| Median (IQR) | 51 (24–104) | 133 (93–202) | [8] | 54 (29–93) | 45 (19–98) | 96 (41–167) | [8] |
| **Any primary care visit in the first 2 months [2]** | | | | | | | |
| N (%) | 8,979 (51.0%) | 1,997 (11.8%) | [8] | 6,339 (52.6%) | 1,443 (53.5%) | 2,888 (28.8%) | [8] |
| **Any primary care visit in year 1 [1]** | | | | | | | |
| N (%) | 15,882 (90.1%) | 13,071 (77.0%) | [8] | 11,423 (94.8%) | 2,389 (88.6%) | 8,250 (82.4%) | [8] |
| **Number of primary care visits per person in year 1[2]** | | | | | | | |
| Mean ± SD | 5.6 ± 5.0 | 3.3 ± 3.6 | [8] | 6.2 ± 5.2 | 4.3 ± 4.2 | 4.0 ± 4.1 | [8] |
| Median (IQR) | 4 (2–8) | 2 (1–5) | [8] | 5 (3–8) | 3 (1–6) | 3 (1–6) | [8] |
| **Any CHC visit in year 1** | | | | | | | |
| N (%) | 2,358 (13.4%) | 464 (2.7%) | [8] | 3,307 (27.4%) | 578 (21.4%) | 520 (5.2%) | [8] |
| **Any visit to an immigrant specialized CHC in year 1[3]** | | | | | | | |
| N (%) | 1,844 (10.5%) | 204 (1.2%) | [8] | 1,231 (10.2%) | 208 (7.7%) | 234 (2.3%) | [8] |
| **Number of CHC visits per person in year 1[4]** | | | | | | | |
| Mean ± SD | 5.0 ± 4.4 | 4.5 ± 3.9 | [8] | 4.2 ± 4.0 | 5.3 ± 4.6 | 4.9 ± 4.3 | [8] |
| Median (IQR) | 4 (2–7) | 3 (2–6) | | 3 (2–6) | 4 (2–7) | 4 (2–6) | [8] |
| **Primary care affiliation assigned at the end of year 1[5] N (%)** | | | | | | | |
| Comprehensive | 7,288 (41.4%) | 6,281 (37.0%) | [8] | 4,474 (37.1%) | 1,069 (39.7%) | 3,999 (39.9%) | [8] |
| Paediatrics | 164 (0.9%) | 99 (0.6%) | | 332 (2.8%) | 46 (1.7%) | 119 (1.2%) | |
| Other primary care | 6,084 (34.5%) | 6,517 (38.4%) | | 3,033 (25.2%) | 711 (26.4%) | 3,532 (35.3%) | |
| CHC, high immigrant/refugee area | 1,017 (5.8%) | 162 (1.0%) | | 953 (7.9%) | 120 (4.5%) | 144 (1.4%) | |
| CHC, specializing in immigrants and refugees | 827 (4.7%) | 42 (0.2%) | | 278 (2.3%) | 88 (3.3%) | 90 (0.9%) | |
| Other CHC | 514 (2.9%) | 260 (1.5%) | | 2,076 (17.2%) | 370 (13.7%) | 286 (2.9%) | |
| No primary care [6] | 1,729 (9.8%) | 3,607 (21.3%) | | 905 (7.5%) | 291 (10.8%) | 1,841 (18.4%) | |
| **Morbidity Burden N (%)** | | | | | | | |
| Any major morbidity [7] | 5,923 (33.6%) | 3,216 (19.0%) | [8] | 3,616 (30.0%) | 684 (25.4%) | 2,354 (23.5%) | [8] |
| Death in the first year | 21 (0.1%) | 23 (0.1%) | | 14 (0.1%) | * | 8 (0.1%) | |

[1] includes BVORs due to small counts.

[2] includes primary care visits to a GP, Pediatrician, or NP; or visits to a GP or NP at a CHC.

[3] Immigrant specialized CHCs includes CHCs located in high immigrant/refugee area and/or specializing in care for immigrants and refugees, includes visits to a GP or NP.

[4] Among those with visits to a CHC, includes visits to a GP or NP.

[5] "Comprehensive" refers to enrollment in any primary care model; "Paediatrics" does not participate in primary care models but does provide primary care; "Other primary care" includes family physicians who are not part of primary care models, usually practicing in a walk-in clinic or as a solo physician; "; "No primary care" refers to having no primary care visits in the previous year.

[6] n = 745 individuals who had no affiliation (i.e., did not have a primary care visit in year 1) were assigned to a primary care model category, likely due to family members attending a primary care visit together and becoming rostered to the same provider, however where the visit was billed as a single visit.

[7] Any major morbidity is defined using the Johns Hopkins ACG® System Aggregated Diagnosis Groups (ADGs) case-mix adjustment system (version 10) which groups diagnostic codes captured in health service use data. Included at least one major ADG characterized as time-limited major; chronic medical, unstable; psychosocial, unstable; progressive or likely to recur; or a malignancy.

[8] statistically significant difference across categories (p<0.05).

*Data are suppressed to reduce the risk of re-identification (for cell sizes <6).

affiliated with a comprehensive or other primary care model by the end of the first year; however privately sponsored refugees were twice as likely to have no primary care affiliation (pre-Syrian era 21.3%; Syrian era 18.4%) compared to government-assisted (pre-Syrian era 9.8%; Syrian era 7.5%) and blended-visa office referred refugees (10.8%). In addition, privately sponsored refugees were much less likely to be affiliated with any CHC model or have any CHC care. For detailed information on primary care and CHC healthcare use, see S6 Table.

## Time to first primary care visit and any CHC visit within first year of landing

Government-assisted (both eras) and blended-visa office referred refugees had their first primary care visit earlier than privately sponsored refugees (both eras) and first primary care visit was earlier for government-assisted and privately sponsored refugees in the Syrian-era compared to their counterparts in the pre-Syrian era (Fig 1). At the end of the first year, 95% of Syrian era government-assisted refugees had a primary care visit compared to 90% of pre-Syrian era government-assisted refugees, 89% of Syrian era blended-visa office referred refugees, 82% of Syrian era privately sponsored refugees, and 77% of pre-Syrian era privately sponsored refugees (p<0.0001) (Table 3).

In comparison to pre-Syrian privately sponsored refugees, time to first primary care visit was significantly earlier for all groups (Fig 2, panel A). Pre-Syrian era government-assisted refugees and Syrian era government-assisted and blended-visa office referred refugees had their

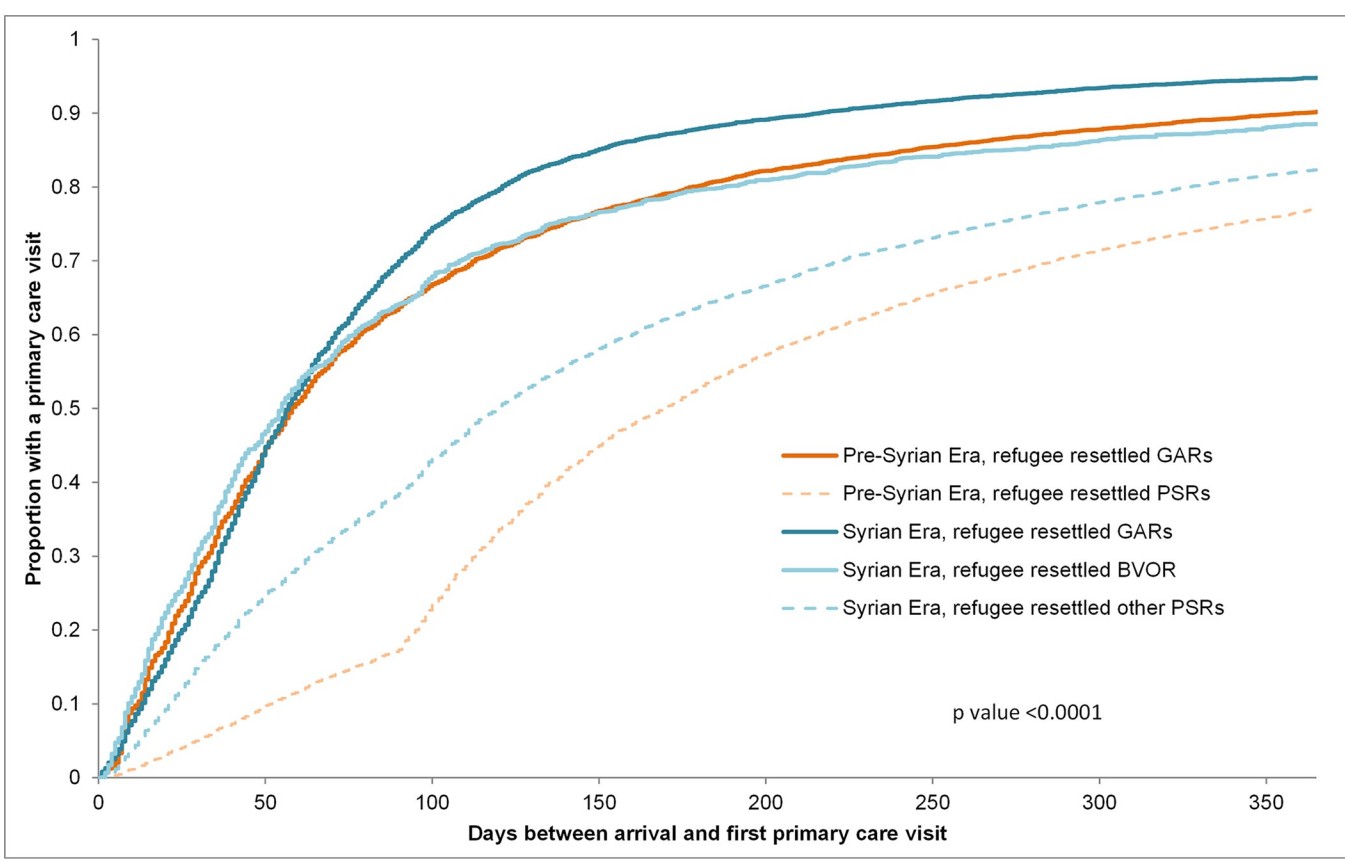

**Fig 1. Kaplan-Meier survival curves for the proportion of resettled refugees having a primary care visit in the first year after landing in Canada, by era of landing and sponsorship model, who landed in Ontario between April 1, 2008 and March 31, 2017.**

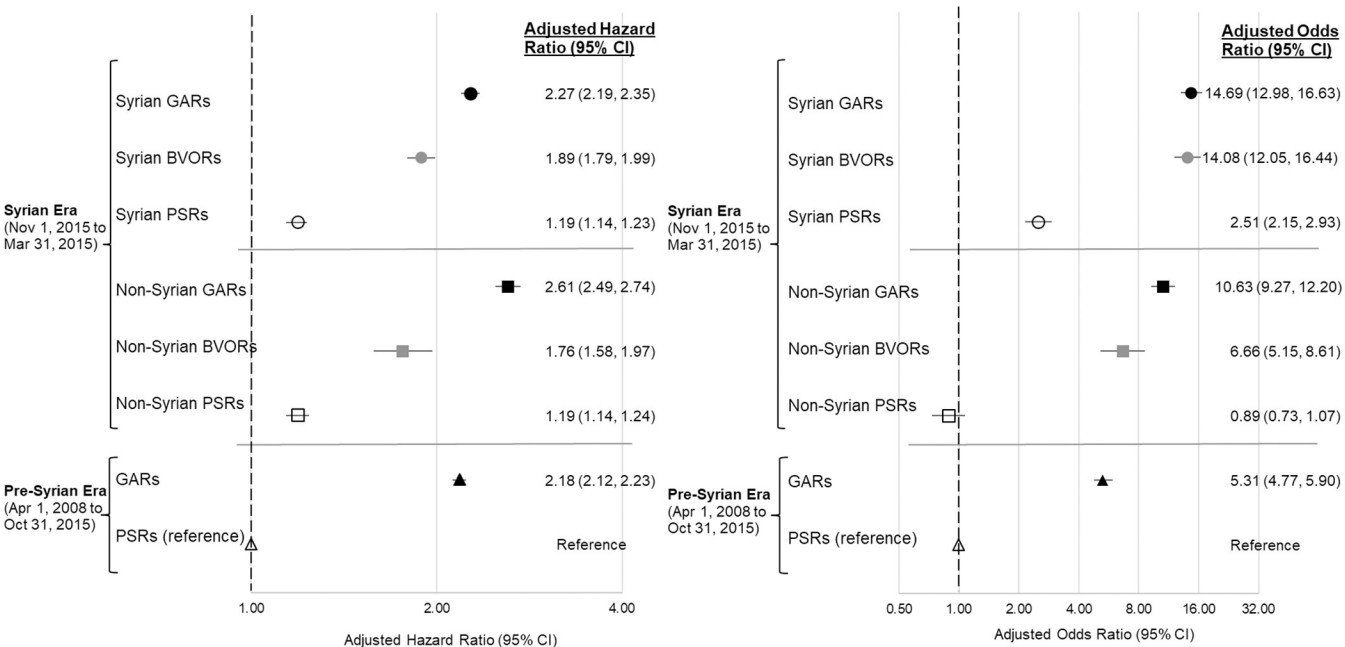

**Fig 2.** The association between sponsorship era + model with time to first primary care visit (adjusted hazard ratio, 95% CI) [panel A] and the odds of a community health centre (CHC) visit (adjusted odds ratio, 95% CI) [panel B] in the first year of resettlement in all resettled refugees who landed in Ontario between April 1, 2008 and March 31, 2017. Adjusted for age group, sex, neighborhood deprivation index, Canadian language ability, world region of citizenship, secondary migration, and season of landing (time to first primary care visit model only) or travel time to a CHC (any CHC visit model only). Reference group is Pre-Syrian privately sponsored refugees.

first primary care visit 2–3 times sooner compared to pre-Syrian era privately sponsored refugees while Syrian era privately sponsored refugees had their first primary care visit slightly earlier. In the Syrian era, non-Syrian government-assisted refugees experienced faster time to first primary care visit compared to Syrian government-assisted refugees but there was no difference between privately sponsored refugees. Syrian era refugees experienced a slight advantage over their pre-Syrian era counterparts in the same sponsorship group. See S7 Table for all full model.

The odds of a CHC visit in the first year were significantly higher for most groups compared to pre-Syrian era privately sponsored refugees (Fig 2, panel B). Pre-Syrian era government-assisted refugees (adjusted odds ratio [aOR] = 5.31, 95% CI 4.77–5.90) and Syrian era government-assisted (Syrian aOR = 14.69, 95% CI 12.98–16.63; Non-Syrian aOR = 10.63, 95% CI 9.27–12.20) and blended-visa office referred refugees (Syrian aOR = 14.08, 95% CI 12.05–16.44; Non-Syrian aOR = 6.66, 95% CI 5.15–8.61) had greater odds of a CHC visit compared to pre-Syrian privately sponsored refugees (p<0.0001). However, Syrian era Syrian privately sponsored refugees had 2.5 times greater odds of a CHC visit compared to pre-Syrian privately sponsored refugees while there was no difference for Syrian era non-Syrian privately sponsored refugees. All Syrian era Syrian government-assisted, blended-visa office referred and privately sponsored refugees had significantly greater odds of a CHC visit compared to their non-Syrian counterparts. Syrian era government-assisted refugees had greater odds of a CHC visit compared to pre-Syrian government-assisted refugees. See S7 Table for full model.

Generally, government-assisted refugees from all countries attended their first primary care visit significantly earlier (Fig 3, panel A) and had higher odds of a CHC visit (Fig 3, panel B)

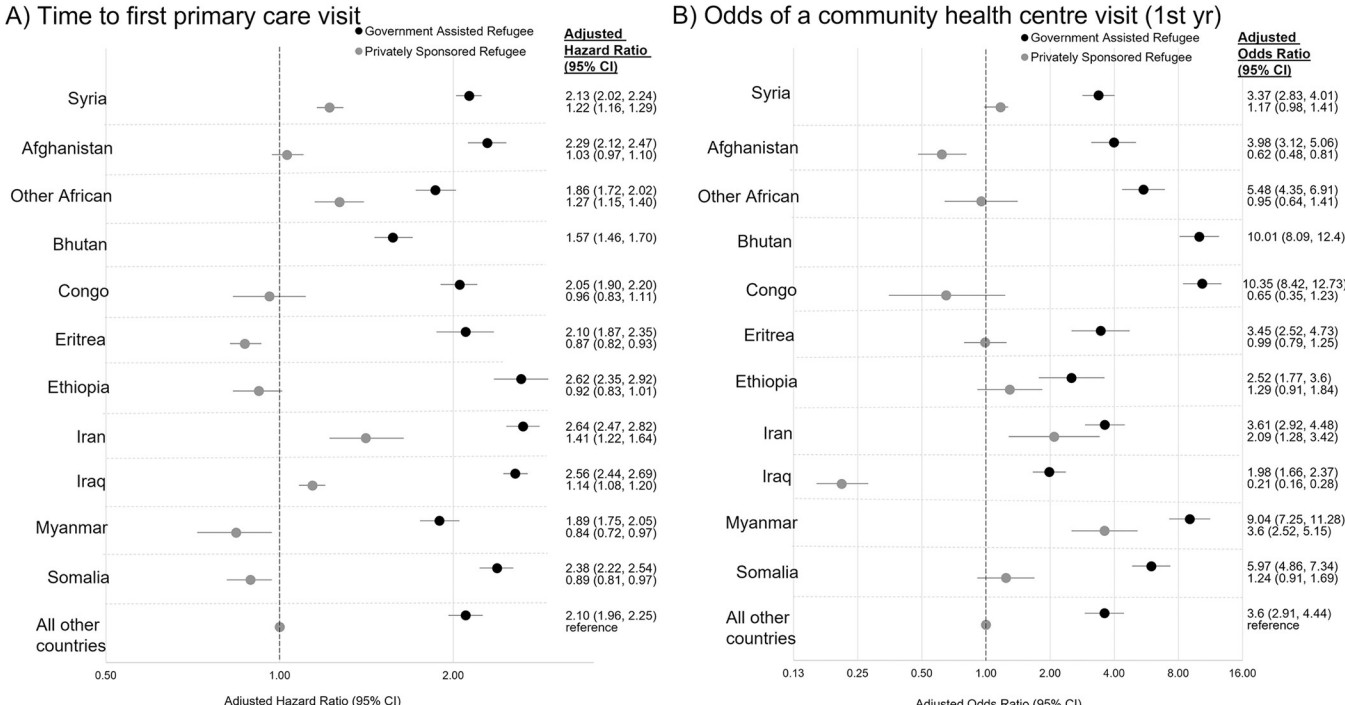

**Fig 3.** The association between country cohort + sponsorship model with time to first primary care visit (adjusted hazard ratio, 95% CI) (panel A) and the odds of a community health centre (CHC) visit (adjusted odds ratio, 95% CI) (panel B) in the first year of resettlement in all resettled refugees who landed in Ontario between April 1, 2008 and March 31, 2017. Adjusted for age group, sex, neighborhood deprivation index, Canadian language ability, secondary migration, landing year and season of landing (time to first primary care visit model only) or travel time to a CHC (any CHC visit model only). Reference group is privately sponsored refugees from all other countries not specified in Figure.

compared to privately sponsored refugees (includes blended-visa office referred refugees) from all other countries not specified (reference group) and in relation to privately sponsored refugees from the same country of origin. Government-assisted refugees from all country cohorts had their first primary care visit 1.5–3 times earlier while privately sponsored refugees from all country cohorts had their first primary care visit between 40% earlier and 16% later compared to privately sponsored refugees from "all other countries". Government-assisted refugees from all country cohorts had between 2–10 times greater odds of a CHC visit while privately sponsored refugees from all country cohorts had between 80% greater and 80% lower odds of a CHC visit compared to the referent group privately sponsored refugees. Of government-assisted refugees and privately sponsored refugees from the same country, there were significant difference in the odds of a CHC visit for all countries except Iran and Ethiopia. Privately sponsored refugees from Iran had their first primary care visit the earliest (adjusted hazard ratio = 1.41 [1.22–1.64]), followed by other Africans, Syrians, and Iraqis, whereas PSRs from Myanmar had greater odds of a CHC visit within the first year (adjusted odds ratio = 3.60 [2.52–5.15]), followed by Iranians. PSRs from Afghanistan and Iraq had lower odds of a CHC visit than the referent group privately sponsored refugees. Time to first primary care visit began steadily decreasing in 2013, preceding the Syrian era (See S8 Table for full model).

## Subgroup analyses

Primary outcome models restricted to those ≥25 years old and adjusted for marital status and education (S9 Table) were similar to adjusted results summarized in Figs 2 and 3. Resettled

refugees with secondary education or less took longer to have their first primary care visit and had lower odds of a CHC visit and those who were single at arrival also took longer to have their first primary care visit.

## Discussion

In this large population-based study of resettled refugees arriving in Ontario, we found the majority of resettled refugees in both the pre-Syrian and Syrian eras had at least one primary care visit in the first year of resettlement; however there was important heterogeneity by resettlement group in many primary care outcomes and morbidity. Privately sponsored refugees were less likely to have a major morbidity compared to government-assisted or blended-visa office referred refugees and took longer to have their first primary care visit, were twice as likely to have no primary care visits and no primary care affiliation and much less likely to have a CHC visit. Our findings demonstrate a modest improvement among privately sponsored refugees and non-Syrian government-assisted refugees in time to first primary care visit in the Syrian era compared to the pre-Syrian era, which may be attributed to greater mobilization and coordination across health and settlement sectors in the Syrian era. The odds of a CHC visit were substantially higher for most resettled refugee groups in the Syrian era, particularly those from Syria. However, we document timelier primary care and use of CHC visits preceding the Syrian era (beginning in 2013). In the examination of multiple cohorts of resettled refugees, we found government-assisted refugees from most countries had more timely primary care and use of CHCs compared to privately sponsored refugees, suggesting that resettlement model and their related selection processes and sponsorship approaches, are a more important and consistent determinant of primary care healthcare use than specific country contexts.

It is recommended that all resettled refugees be seen by a primary care provider soon after arrival [37]. Differences in primary health care use between resettled refugee groups can likely be attributed to several factors related to both selection criteria and sponsorship approaches. Lower health care use amongst privately sponsored refugees may be related to the fact that they are chosen by sponsors rather than based on urgent resettlement need or vulnerability [20], and therefore may be less likely to have health concerns. Competing resettlement priorities such as early employment, (also facilitated by selection factors–family job networks, greater knowledge of English and higher education [14, 19]) may delay accessing primary care. Lower healthcare use could also be related to the challenges sponsors may face in effectively navigating primary care [38]. The government-assisted refugee program selects for refugees with greater vulnerability including health care need, and this may drive the use of healthcare services. Given this important dissimilarity between government-assisted refugees and privately sponsored refugees, the comparison to blended-visa office referred refugees is instructive. They are chosen similarly to government-assisted refugees but resettled differently and while both privately sponsored and blended-visa office referred refugees are resettled by private citizens, blended-visa office referred refugees sponsors are entirely civic volunteers rather than family members. Comparing blended-visa office referred to government-assisted refugees directly was not done analytically due to the recency of the blended-visa office referred refugee program and the small population. However, Fig 2 panel A suggests that likely greater health care needs (due to selection) among both government-assisted refugees and blended-visa office referred refugees may have been better met by the formal resettlement services available to government-assisted refugees given faster time to first primary care visit among government-assisted refugees compared to blended-visa office referred refugees.

Despite the relatively large number of resettled refugees arriving in a short period in the Syrian era and no change in overall CHC capacity, there was a dramatic increase in use of CHCs. Our findings provide support for the statement that the CHC sector specifically mobilized to provide support and meet the needs of an influx of resettled refugees [6]. Further work should explore the longer-term impact of the use of this interdisciplinary model of care for refugees.

In general, studies have shown that refugees in Ontario are less likely to receive preventive health care measures [33, 39–43] but that immigrants enrolled in CHCs have better preventive health care (e.g., cancer screening) and lower emergency department visits compared to other Ontario residents [44]. Several studies were identified related to health care use of recently resettled refugees stratified by resettlement group however none had sample sizes that would be considered generalizable to these respective populations. A rapid evaluation of the Syrian resettlement effort conducted by the federal immigration agency [45] suggested that more privately sponsored refugees (64%) than government-assisted refugees (39%) received "help finding a doctor on their own" and more Syrian privately sponsored refugees (85%) than Syrian government-assisted refugees (71%) were taught "how to get healthcare". Three cross-sectional studies using primary data included 400 Syrian refugees landing in 2015–2016 in the Greater Toronto Area [46–48] including privately sponsored refugees (52%), government-assisted refugees (44%) but only a small number of blended-visa office referred refugees (3%), limiting conclusions specific to blended-visa office referred refugees. These studies found that 51% of refugees perceived their physical and mental health to be similar to a year earlier, 33% reported better health and 16.5% reported worse health. Over half reported unmet healthcare needs citing long wait times, service costs or lack of time to seek care as the top 3 reasons for unmet need [46]. Oda et al., 2019 [47] reported government-assisted refugees had significantly lower perceived physical and mental health and higher unmet health care needs than privately sponsored refugees and Tuck et al., 2019 [48] reported that similar trends in unmet healthcare needs persisted 6 months to a year after arrival.

Other integration outcomes provide some context for our findings. In a 2016 Immigration, Refugees and Citizenship Canada evaluation of employment outcomes found unemployment was higher initially amongst government-assisted refugees, but improved over time with incomes converging with privately sponsored refugees after 10 years [19]. These findings were largely consistent with findings from a recent peer-reviewed study [49]. It is posited that this trajectory is related to government-assisted refugees accessing official-language and employment training related services [16] while privately sponsored refugees were motivated to offset sponsorship costs (often borne by relatives) and prioritized early employment instead of seeking employment counseling from resettlement agencies. The need for formal resettlement training for sponsors [18] and/or the sharing of advice from experienced sponsorship groups in navigating settlement services [50] has been suggested to improve longer-term employment outcomes for privately sponsored refugees. We speculate that these circumstances may be relevant for healthcare navigation and access for privately sponsored and blended-visa office referred refugees, particularly so for the latter considering the possibility of blended-visa office referred refugee's greater health care needs due to selection factors.

## Limitations

Our study has limitations. In the first few months of the Syrian era, informal primary health care offered soon after arrival (primarily catch-up vaccinations and urgent dental care) [51] and federally insured supplemental services are not included in the databases used for this study. There are a small number of formal primary health care centres other than CHCs

dedicated to refugee health care; these centres could not be separately identified and examined, although most focus on asylum-seekers without provincial healthcare insurance. We restricted analyses to residents of urban areas and therefore cannot describe primary care and CHC use for the very small numbers of refugees resettled in rural areas, most of whom were privately sponsored refugees. Drive time to the nearest CHC may not accurately capture the practical effort and time needed to overcome longer distances (e.g., time consuming transportation by bus or lack of public transportation altogether) and thus residual confounding may remain. There is acknowledged heterogeneity among privately sponsored refugee sub-groups, both in terms of whether refugees are related to sponsors as well as selection factors shaping healthcare need; however these groups were not analysed separately due to small sample sizes. We were unable to shed light on the quality of health care received by resettled refugees using administrative databases; however qualitative research with Syrian government-assisted refugees [52] and African government-assisted and privately sponsored refugees [38] suggests room for improvement. Finally since primary health care reform in Ontario has been ongoing for the past 20 years [53] and no population-level group was included in our analyses, it is uncertain whether some of the improvements in primary care outcomes in the Syrian era also be partially attributed to wider changes to primary care.

## Conclusions

Most resettled refugees in Ontario engaged with the primary health care system in the first year after arrival, with improvement during the Syrian crisis as Canada undertook its largest refugee resettlement effort to date. Lower morbidity and less timely primary care access amongst privately sponsored refugees compared to government-assisted/blended-visa office referred refugees in the first year after arrival can likely be attributed to selection factors. However, greater proportions of privately sponsored refugees unaffiliated to any primary care model after the first year and with much lower use of specific healthcare models geared towards new immigrants, may signal gaps in sponsors' knowledge of available healthcare services. Primary care use amongst blended-visa office referred refugees suggests some unmet needs and that their sponsors may also benefit from training. More research is needed to understand outcomes for blended-visa office referred refugees who are selected based on vulnerability (like government-assisted refugees) and likely to have greater medical and social needs but resettled by citizen sponsors who are unknown to them, particularly given the promotion of similar sponsorship models in other countries.

## Supporting information

**S1 Fig. Database linkage and cohort selection flowchart.**
(DOCX)

**S1 Table. Study data sources.**
(DOCX)

**S2 Table. List of study variables.**
(DOCX)

**S3 Table. Dataset creation and analysis plan.**
(DOCX)

**S4 Table. List of all primary care (PC) visit feecodes.**
(DOCX)

**S5 Table. List of all Ontario Community Health Centres (CHCs), as well as indicators for those who specialize in care for immigrants/refugees and those who are located in high immigrant/refugee areas.**
(DOCX)

**S6 Table. Other healthcare use in the first year among resettled refugees that landed in Ontario between April 1, 2008 and March 31, 2017, by era of landing and sponsorship model.**
(DOCX)

**S7 Table. Cox proportional hazard ratios for the association between sponsorship model and era on time to first primary care (PC) visit and logistic odds ratio for the association between sponsorship era and era on any community health centre (CHC) visit in the first year of resettlement in all resettled refugees who landed in Ontario between April 1, 2008 and March 31, 2017.**
(DOCX)

**S8 Table. Cox proportional hazards ratios for the association between sponsorship model and country cohort on time to first primary care (PC) and logistic odds ratios for the association between sponsorship model and country cohort on any community health centre (CHC) visit in the first year of resettlement in all resettled refugees who landed in Ontario between April 1, 2008 and March 31, 2017 including adjustment for landing year.**
(DOCX)

**S9 Table. Cox proportional hazards estimates of the association between sponsorship model and era on primary care (PC) and logistic odds ratio for any community health centre (CHC) visit in the first year of resettlement in all resettled refugees who landed in Ontario between April 1, 2008 and March 31, 2017 among those aged ≥ 25.**
(DOCX)

## Acknowledgments

Parts of this material are based on data and information compiled and provided by the Canadian Institute for Health Information (CIHI) and Immigration, Refugees Citizenship Canada (IRCC). However, the analyses, conclusions, opinions and statements expressed herein are those of the authors, and not necessarily those of CIHI or IRCC. The authors wish to thank Duncan Lawrence, Jens Hainmueller and Jeremy Weinstein of the Stanford Immigration Policy Lab for their expert comments on a draft of the manuscript and interpretation of findings.

## Author Contributions

**Conceptualization:** Susitha Wanigaratne, Jennifer Rayner, Richard H. Glazier, Anja Kilibarda, Astrid Guttmann.

**Formal analysis:** Hong Lu.

**Funding acquisition:** Astrid Guttmann.

**Methodology:** Therese A. Stukel, Hong Lu, Astrid Guttmann.

**Project administration:** Sima Gandhi, Astrid Guttmann.

**Supervision:** Astrid Guttmann.

**Writing – original draft:** Susitha Wanigaratne, Sima Gandhi, Astrid Guttmann.

**Writing – review & editing:** Susitha Wanigaratne, Jennifer Rayner, Richard H. Glazier, Therese A. Stukel, Hong Lu, Sima Gandhi, Natasha R. Saunders, Michaela Hynie, Anja Kilibarda, Astrid Guttmann.

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
