## [Decision Letter · Decision Letter 0]

30 Mar 2023

PONE-D-22-26649Association of Canada’s unique refugee sponsorship model and primary health care use in the 1st year of arrival: a population-based cohort studyPLOS ONE

Dear Dr. Wanigaratne,

Thank you for submitting your manuscript to PLOS ONE. After careful consideration, we feel that it has merit but does not fully meet PLOS ONE’s publication criteria as it currently stands. Therefore, we invite you to submit a revised version of the manuscript that addresses the points raised during the review process. Reviewers have requested minor clarifications within the manuscript and a final revision for clarity of writing. No significant revision of methods, interpretation, or reporting is required. Please submit your revised manuscript by May 14 2023 11:59PM. If you will need more time than this to complete your revisions, please reply to this message or contact the journal office at plosone@plos.org. Please include the following items when submitting your revised manuscript:A rebuttal letter that responds to each point raised by the academic editor and reviewer(s). You should upload this letter as a separate file labeled 'Response to Reviewers'.A marked-up copy of your manuscript that highlights changes made to the original version. You should upload this as a separate file labeled 'Revised Manuscript with Track Changes'.An unmarked version of your revised paper without tracked changes. You should upload this as a separate file labeled 'Manuscript'.If applicable, we recommend that you deposit your laboratory protocols in protocols.io to enhance the reproducibility of your results. Protocols.io assigns your protocol its own identifier (DOI) so that it can be cited independently in the future. For instructions see: https://journals.plos.org/plosone/s/submission-guidelines#loc-laboratory-protocols. Additionally, PLOS ONE offers an option for publishing peer-reviewed Lab Protocol articles, which describe protocols hosted on protocols.io. Read more information on sharing protocols at https://plos.org/protocols?utm_medium=editorial-email&utm_source=authorletters&utm_campaign=protocols.

We look forward to receiving your revised manuscript.

Kind regards,

Blake Byron Walker, Ph.D.

Academic Editor

PLOS ONE

Journal Requirements:

3. PLOS requires an ORCID iD for the corresponding author in Editorial Manager on papers submitted after December 6th, 2016. Please ensure that you have an ORCID iD and that it is validated in Editorial Manager. To do this, go to ‘Update my Information’ (in the upper left-hand corner of the main menu), and click on the Fetch/Validate link next to the ORCID field. This will take you to the ORCID site and allow you to create a new iD or authenticate a pre-existing iD in Editorial Manager. Please see the following video for instructions on linking an ORCID iD to your Editorial Manager account: https://www.youtube.com/watch?v=_xcclfuvtxQ"

4. Please update your submission to use the PLOS LaTeX template. The template and more information on our requirements for LaTeX submissions can be found at http://journals.plos.org/plosone/s/latex.

Reviewers' comments:

Reviewer's Responses to Questions

**Comments to the Author**

1. Is the manuscript technically sound, and do the data support the conclusions?

Reviewer #1: Partly

Reviewer #2: Yes

2. Has the statistical analysis been performed appropriately and rigorously? 

Reviewer #1: Yes

Reviewer #2: Yes

3. Have the authors made all data underlying the findings in their manuscript fully available?

Reviewer #1: Yes

Reviewer #2: Yes

4. Is the manuscript presented in an intelligible fashion and written in standard English?

Reviewer #1: Yes

Reviewer #2: Yes

5. Review Comments to the Author

Reviewer #1: Thank you for the invitation to review this work. I have accepted this invitation mainly out of curiosity, not being an expert in the field, but being an involved world citizen, and general practitioner working in the Netherlands, where I encounter different problems with the access to care of refugees. So this is also my disclaimer. I do, however, also have methodological skills as an epidemiologist with which I give my feedback to this manuscript. I have not encountered methodological major issues, but only minor comments. In general, the manuscript is very long due to the details shared. The advantage of this, is the transparancy, but the disadvantage is the difficulty to remain focussed while reading.

Next, I feel that the conclusions drawn (as described in the abstract) go beyond the scope of this article. Specifically, as it is unknown what the content of the consultations with the CHC were.

In the introductions, authors clearly explain how the health care system in Canada is organised, both in general terms, as more specific for the demands of refugees. The rationale is clear and focusses in the differences in navigating to the health care system between refugees taking part in the different forms of resettlement programs.

Although abbreviations are explained, the use of many abbreviations (PSR BVOR GARS) is a bit confusing for an outsider.

At the end of the introduction, two hypothesis are mentioned. The first is clear and explained, but the second remains unclear to me, as I don’t truly see why the authors mean with “.., related to the higher profile government, civic and professional response to resettling those fleeing the Syrian conflict” (line 137-8) Please elaborate a bit more to keep the reader focussed.

In the methods section, authors provide detailed information on the data sources, and define “linked and unlinked refugees” meaning (if my interpretation is correct) that refugees were not included in the RPDB (unlinked refugees). It is unclear WHEN people are registered in that database. Is a direct contact with a HCP obligatory for that? So, are the unlinked refugees people who haven’t had a single contact in the study period?

It could be helpful to provide a Figure clarifying how data bases were combined. It seems that Figure 1 in the supplementary files have less databases mentioned than the method section.

All statistics are explained in detail. Please acknowledge that due to the very large number of patients included in the database, the change of having a statistically significant difference for any comparison is large. The clinical relevance of difference needs more attention. Here, authors have chosen to present Odds ratios (line 324-332) where relative risk ratios would be more appropriate, taken into consideration the very high prevalence of a CHC visit in this population.

In the discussion, at some points I don’t follow the argumentation of the authors. For example, in line 369-370, authors mention “While morbidity was lower among PSRs, lower primary care use in the first year of resettlement may have important implications for prevention, quality of primary care and longer-term adverse health outcomes.” As a GP I disagree with that statement. Why would GPs start preventative measures when a patient visits the practice, e.g. for headache or a skin problem. This suggestions goes beyond the scope of this study, and I would suggest the authors to refrain from expressing this kind of unsupported thoughts.

Reviewer #2: Title

- Associations "between" not "of", as the latter implies that the object of study is a formal administrative structure between the two, whilst the former implies a statistical association.

- Replace numerical "1st" with "first"

- Maybe a reformulation would be more clear. Something along the lines of "Primary health care utilization amongst refugees in their first year in Canada: a population-based cohort study"

Keywords

- some of the selected keywords are a bit unusual and/or non-specific, e.g., "centres, community health". The authors are encouraged to reconsider these carefully.

Formalities

- In-text citation is inconsistent (e.g., ln 66 & ln 68). Please double-check for formatting, also in the references list.

- There are a few writing errors scattered throughout, e.g., ln 87 comma splice. The authors should conduct a thorough final revision for writing.

High-level comments

- The writing is, at times, unclear. The authors are strongly advised to consider revising the manuscript for clarity.

- Is the Canadian mixed sponsorship model entirely unique? I was under the impression that a mixed model is also used in some European countries, but I might be wrong. If there are any other models that are somewhat similar, e.g., following the influx of refugees fleeing from Ukraine into the EU/EEA, these should be differentiated in the first paragraph of the background section.

- Experimental setup is very well designed and executed, including selection of criteria/variables.

- Correct selection and use of statistical tests.

- Results are presented clearly and logically and the tables provide useful summaries of the results.

- Interpretation of results is clear and does not meander into unqualified speculation.

- Study limitations appropriately reflected.

Other comments

- Table 1 is very useful. The authors should refer to it earlier in the text. The footnotes for Table 1 should be seperated by line breaks.

- Background section very informative!

- Background ln 128, indicate what is defined as the pre-Syrian period. April 1, 2008 to Nov 2015?

- Paragraph starting on ln 144: indicate the number of excluded cases for each exclusion criterion!

- Para. starting on ln 171: please define "era of landing" as a binary variable and explain in the following sentence that it pertains to the war in Syria. The reasoning for the exact dates will need to be provided.

- ln 207, are adults defined as >18 or >=18 years of age? In most cases, 18 year-olds are considered legal adults in Canada, to my knowledge.

- Did the authors use census neighbourhoods (do these exist?) or census dissemination areas (DAs)? They look to me like DAs.

- Replace instances of 1st with first in text, e.g., ln 296.

_ I wonder to what degree in-community health care (i.e., without interfacing with the public health care system in Ontario) plays a role and how this might differ between various subpopulations?

-

6. PLOS authors have the option to publish the peer review history of their article (what does this mean?). If published, this will include your full peer review and any attached files.

Reviewer #1: **Yes: **Marco H. Blanker, MD PhD, associate professor, general practitioner and epidemiologist.

Reviewer #2: No

---

## [Author Response · Author response to Decision Letter 0]

16 May 2023

Please see separate document outlining our responses to the reviewer comments. Thank you!

---

## [Editor Report · Decision Letter 1]

7 Jun 2023

Primary health care utilization in the first year after arrival by refugee sponsorship model in Ontario, Canada: A population-based cohort study

PONE-D-22-26649R1

Dear Dr. Wanigaratne,

We’re pleased to inform you that your manuscript has been judged scientifically suitable for publication and will be formally accepted for publication once it meets all outstanding technical requirements.

Kind regards,

Blake Byron Walker, Ph.D.

Academic Editor

PLOS ONE
---

## [Editor Report · Acceptance letter]

18 Jul 2023

PONE-D-22-26649R1 

Primary health care utilization in the first year after arrival by refugee sponsorship model in Ontario, Canada: A population-based cohort study 

Dear Dr. Wanigaratne:

I'm pleased to inform you that your manuscript has been deemed suitable for publication in PLOS ONE. Congratulations! Your manuscript is now with our production department. 

Kind regards, 

on behalf of

Prof. Dr. Blake Byron Walker 

Academic Editor

PLOS ONE